**communications** engineering

# Multi-channel ultrasonic Bessel vortex beams by spatial multiplexing metalens
Yinjie Su, Di Wang, Zhongming Gu ✉, Chen Liu & Jie Zhu ✉

The generation of acoustic vortexes sparks intense research interest since they have applications in modern wave-based technologies, such as underwater communication and particle manipulation. However, the existing schemes mainly rely on a phase mask to excite a single vortex beam, thereby lacking the functionality and adaptability for practical scenarios. In this article, we propose a feasible methodology to realize multi-channel ultrasonic Bessel vortex beams at megahertz. By leveraging the concept of spatial multiplexing, the adjacent pixel of the metalens can be assigned to independently generate non-diffraction ultrasonic vortices with different topological charge and spatial orientation, without losing the characteristics of the helicoidal wavefront. We experimentally designed a four-channel metalens with a high fabrication accuracy of 0.2 mm pixel size and measured the far-field ultrasound distribution in the water. Both topological charge and radiation direction of the generated vortices can be precisely controlled as predicted, showcasing great agreement with simulation results with a directional error of less than 1°. Moreover, the intensity of the vortex can be tuned by gradually combining multiple channels into one. The proposed scheme enhances the flexibility of manipulating ultrasonic vortex and offers more possibilities in designing multi-functional ultrasound devices.

There are many interesting fluidic phenomena with vortex structures in nature, like tornado, whirlpool and backwash. As a counterpart in wave system, the acoustic vortex can also hold the profile of vortex geometry by imposing circumferential phase gradient to the wavefront, in which the acoustic wave travels in a corkscrew way along the propagation direction[1-8]. Recently, the acoustic vortex has been widely studied due to its unique properties, such as orbital angular momentum (OAM) and the singularity of zero intensity. It also opens up a new degree of freedom for wave-matter interaction, inspiring a series of unexpected functionalities. The donut-shaped intensity distribution can trap the particles at the vortex center[9-14], meanwhile the spinning properties stemming from the OAM can induce a torque on the illuminated object, leading to a rotational motion[15-23]. Moreover, the acoustic vortices with different orders are topologically distinct and orthogonal in the Hilbert space, thus can be encoded for high-capacity communications[24-26]. The acoustic vortex can also demonstrate the rotational Doppler phenomenon, generating frequency shifts upon coupling with rotating acoustic sources or metasurfaces[27,28]. This effect broadens the parameter space for information encoding[29].

In recent years, many efforts have been devoted to study the generation of acoustic vortices. These methods can be broadly categorized into three technical routes based on their operational principles and flexibility. The first route relies on active transducer arrays[30,31], which offer dynamic programmability for vortex generation but at the cost of complex electronic control systems and high power consumption. The second route utilizes traditional passive components, such as spiral phase plates[32] or optoacoustic conversion elements[33], which provide a compact and simple solution yet suffer from inherent limitations in functionality reconfiguration and single-operation mode. The third route is built upon acoustic metamaterials and metasurfaces, whose unique properties have transformed wavefront manipulation, enabling precise control over acoustic fields at the sub-wavelength scale[34-43]. Notably, this route has further branched out into advanced design strategies, including topology-optimized metasurfaces for broadband vortex generation[38] and non-Hermitian metamaterials for switchable vortex emission[39,40]. While metasurfaces have demonstrated remarkable capabilities in wavefront manipulation, such as focusing[44-46] and holography[47-49] in the MHz band, most existing passive designs are confined to generating a single vortex beam or a static multifunctional field. Consequently, the efficient generation of multichannel and multidirectional vortex beams in water via a single, passive device remains an open challenge, primarily due to the difficulty in decoupling multiple spatial information channels.

In this study, a multi-channel ultrasonic metalens based on spatial multiplexing is proposed, aiming for the efficient and flexible creation of ultrasonic vortex beams. By predesigning the pixel arrangement of the

Institute of Acoustics, School of Physics Science and Engineering, Tongji University, Shanghai, People's Republic of China.
✉e-mail: zhmgu@tongji.edu.cn; jiezhu@tongji.edu.cn

metalens, multiple channel information is encoded into one design, enabling the conversion of incident plane waves into four independent ultrasonic vortex beams. These channels, bearing either identical or distinct topological charges and spatial orientations simultaneously, are validated through simulations and experiments, demonstrating the versatile multiplexing capability of the metalens. Moreover, by modifying the pixel arrangement strategy to construct a two-channel functionality, the intensity of ultrasound vortex beams can be tuned on demand. The proposed design overcomes the limitations of previous schemes that are confined to single-function regulation, and provides an approach for multi-channel and multi-functional ultrasound manipulation.

## Results

### Design Principle of spatial multiplexing metalens

To enable flexible and efficient generation of underwater ultrasonic vortex beams, we propose a multiplexing metalens capable of simultaneously producing multiple vortex beams that can be either collectively or selectively activated according to operational requirements, thereby enhancing the utilization efficiency of a single device. As schematically illustrated in Fig. 1a, the proposed metalens converts an incident plane wave into four decoupled vortex beams through precisely engineered phase modulation.

Here, $m$ denotes an acoustic field incorporating a helical phase term $\exp(jm\theta)$, where $\theta$ represents the azimuthal coordinate. During axial propagation, such a vortex exhibits $m \times 2\pi$ phase variation around the central phase singularity in the transverse cross-sectional plane. Consider the pure acoustic vortex suffers from severe diffraction effects leading to rapid energy divergence along propagation distance. We introduce a Bessel phase modulation strategy for constructing non-diffracting vortex beams, see Supplementary Information Note 1 for the modulation of Bessel sound beam. Furthermore, spatial decoupling of the vortex beams is achieved through directional beam steering, where each channel propagates along different axis lines. The phase profile for individual channels can be expressed as:

$$\varphi_i = m_i\theta + k_0 \sin\alpha \sqrt{x^2 + y^2} - k_0\left(x\cos\beta_i + y\cos\gamma_i\right) \quad (1)$$

where $k_0 = 2\pi f/c_0$ represents the wavenumber in water, and $f, c_0$ denote the ultrasonic frequency and the sound speed in water, respectively. The azimuthal angle $\theta$ is defined by $\tan\theta = y/x$. Parameter $\alpha$ denotes the base angle required for generating ultrasonic Bessel beams. $\beta$ and $\gamma$ correspond to the angles between the beam propagation axis and the positive $x$-axis and $y$-axis, respectively. The operating frequency is set to 2 MHz in the following simulations and experiments, which is a widely-adopted frequency in many fields of ultrasonic engineering. The parameter $\alpha$ which governs both the interference intensity of Bessel vortex beams and their non-diffracting propagation range, exhibits a positive correlation with beam intensity but a negative correlation with effective propagation distance. To achieve optimal balance, $\alpha$ is uniformly set to 12° through systematic parametric optimization. Meanwhile, all four vortex beams are initially designed with deflection angles 12° relative to the $z$-axis.

The design methodology to construct the phase array can be described by the concept of spatial multiplexing. Typically, a metalens can be considered as an N×N array of pixels, with each pixel having a size of a, as shown in Fig. 1b. When encoding the metalens pixels using the spatial multiplexing scheme, adjacent pixels are assigned to different channels. For example, the yellow pixel as shown in Fig. 1b is encoded for Channel 1, while its three neighboring pixels are encoded for Channels 2-4 respectively. Pixels belonging to the same channel are no longer arranged adjacently but are instead spaced apart from each other through interleaving with pixels from other channels.

The effectiveness of this metalens can be analyzed theoretically by euqalfrequency curves. Figure 1c displays the k-space range of signals encoded by the metasurface before and after spatial multiplexing, as denoted by the gray region, called Spatial Nyquist Region. The blue region represents the Cut-off Wavenumber Circle, determined by the wavelength of incident wave. Signals beyond this circle become evanescent waves, thus can hardly reconstruct the image in the far field. If the lattice constant is extended, i.e., a = 0.2 mm after the spatial multiplexing, the Spatial Nyquist Region of signals encoded by each channel still covers most of the Cut-off Wavenumber Circle in k-space. This demonstrates that in real space, each channel of the spatially multiplexing metalens can reconstruct the majority of the ultrasonic field at 2 MHz, with a slight portion of high-frequency components being lost.

In the following simulations and experiments, the metalens designed to generate multi-channel ultrasonic vortex comprises 200 × 200 pixels, with each pixel dimension set to 0.2 mm. Figure 2a demonstrates the phase distributions of four downsampled channels (all configured with +1-order vortices) generated through Eq. (1), while Fig. 2b displays the corresponding magnified view of Channel 4. The spatial multiplexing scheme encodes these phase patterns into the metalens through parity-dependent pixel allocation. Channel 1 occupies Odd-numbered rows and columns; Channel 2 occupies Odd-numbered rows, even-numbered columns; Channel 3

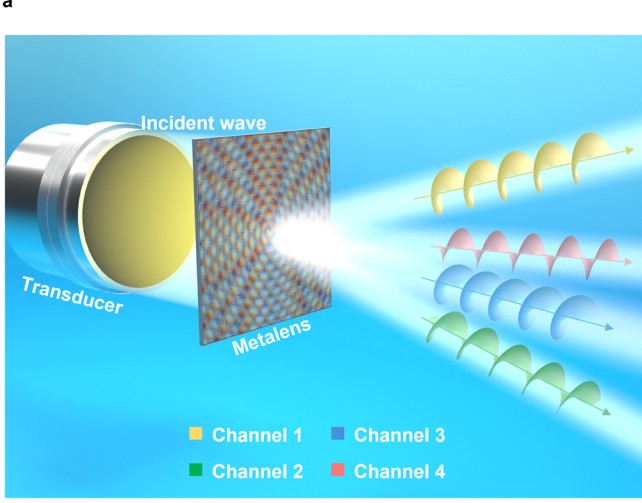

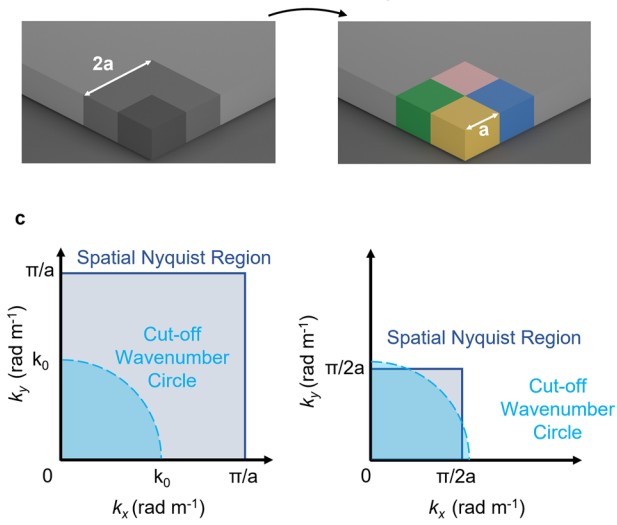

**Fig. 1 | Design principles of spatial multiplexing metalens. a** The metalens converts a plane wave into four acoustic vortices in different directions. **b** Before and after applying the space multiplexing scheme, the pixel distribution of the metalens. **c** The k-space encodable signal range of the full metalens (left) compared with its spatially multiplexed single-channel counterpart (right).

**Fig. 2 | Phase distribution required to generate multi-channel acoustic vortex. a** Phase distribution of different channels encoded acoustic vortex towards different directions. **b** A zoom-in view of the phase distribution of Channel 4. **c** Phase distribution of the metalens including all four channels. **d** Each pixel is designed as a pillar of different heights. The metalens converts plane wave into four acoustic vortices in different directions. All phase maps are plotted using the same colorbar ($[0, 2\pi]$).

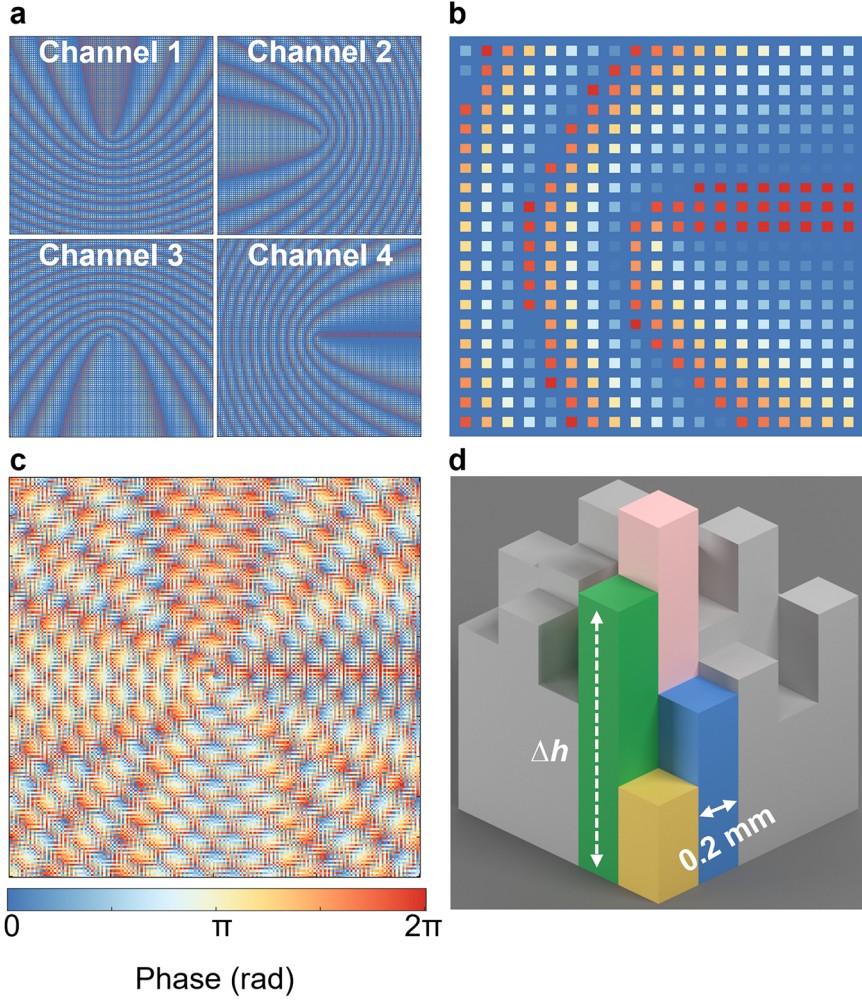

occupies Even-numbered rows, odd-numbered columns and Channel 4 occupies Even-numbered rows and columns.

The final phase distribution of the metalens is shown in Fig. 2c. It should be emphasized that our approach is essentially different from the method of directly superposing multiple phase modulations, see Note 2 in Supplementary Information for detailed discussion.

Each pixel is geometrically modeled as a pillar structure. The phase shift induced by acoustic wave transmission through these height-modulated pillar is determined by:

$$(k_0 - k_1)\Delta h(x, y) = \varphi(x, y) \qquad (2)$$

where $k_1 = 2\pi f/c_1$ and $c_1 = 2200 \text{ m s}^{-1}$ represent the wavenumber and sound speed in the solid material, respectively. The term $\Delta h(x, y)$ denotes the height variation of the rectangular pillars, as shown in Fig. 2d. To prevent structural deformation caused by excessively thin profiles, a base thickness $h_0 = 1mm$ is incorporated into the design. Consequently, the total thickness distribution of the metalens is governed by $h(x, y) = h_0 + \Delta h(x, y)$.

## Performance of metalenses with $+ 1$ order vortex in each direction

Firstly, the three-dimensional acoustic intensity distribution generated by the metalens in the far field has been simulated numerically, as shown in Fig. 3a. Figure 3b displays the beam directivity patterns of Channels 3 and 4 in the $yoz$ plane, confirming that these vortex beams propagate along a predetermined direction. The distribution of sound intensity in the $xoz$ plane ($y = 0$) is illustrated in Fig. 3c. It is observed, with increasing propagation

distance, the vortex beams diverge spatially along distinct trajectories, enabling clear signature of the vortex beams associated with two separate channels. It is noteworthy that the generation of sidelobes is an inherent consequence of the Bessel phase modulation applied to the vortices.

Figure 3d illustrates the $xoy$ plane of the acoustic intensity at a distance of $z = 62.0$ mm away from the metalens. The vortex beams corresponding to the four channels are denoted by white dashed boxes, showing the majority of the acoustic energy is concentrated within the desired vortex beams. The central singularities of the four vortices are positioned approximately 13.5 mm from the center of the ultrasonic field, indicating that the tilt angle of each vortex beam is approximately 12.3°. This ultrasound distribution persuasively verifies the effectiveness of the proposed methodology. For clarity of observation, we have exclusively extracted the phase distribution corresponding to the ultrasonic field within the white frame, as illustrated in Fig. 3e. The resulting phase maps demonstrate that each observed vortex possesses a $+ 1$ topological charge. The observed phase inhomogeneity is attributed to the inclination of the ultrasonic beams relative to the $xoy$ plane, suggesting that the resultant phase can be interpreted as a superposition of the conventional vortex phase and a phase component associated with oblique propagation.

The experimental validations are also conducted for better examination of the designed ultrasound vortices. Figure 4a and b presents the fabricated prototype of the proposed metalens and the corresponding experimental setup, respectively. The sample is fabricated following the aforementioned method, with an additional frame incorporated to ensure stable fixation on the transducer, maintaining their relative positions. The ultrasonic field modulated by the metalens is measured using a hydrophone

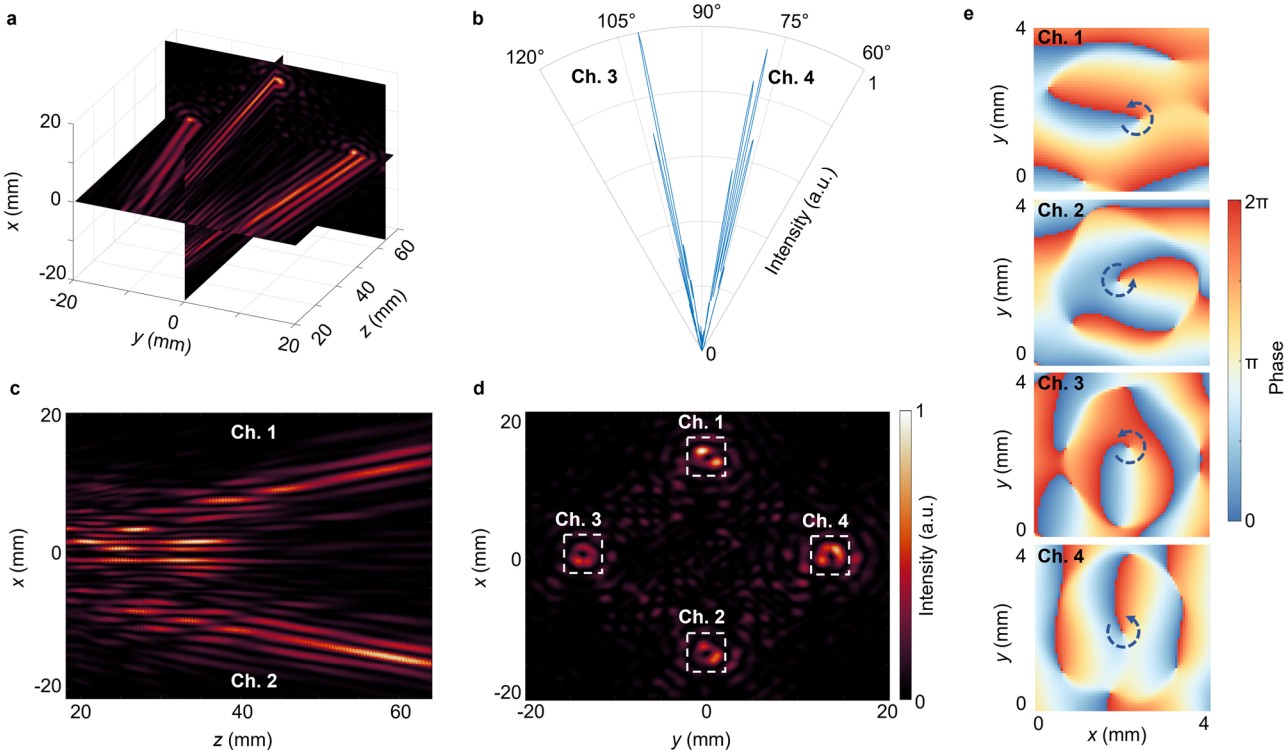

**Fig. 3 | Simulation results of multi-channel acoustic vortex. a** 3D cross-section of the entire sound field. **b** Directivity in the *y* = 0 plane. **c**, **d** Intensity distribution in *xoz* plane and *xoy* plane. **e** Phase distribution of each acoustic vortex.

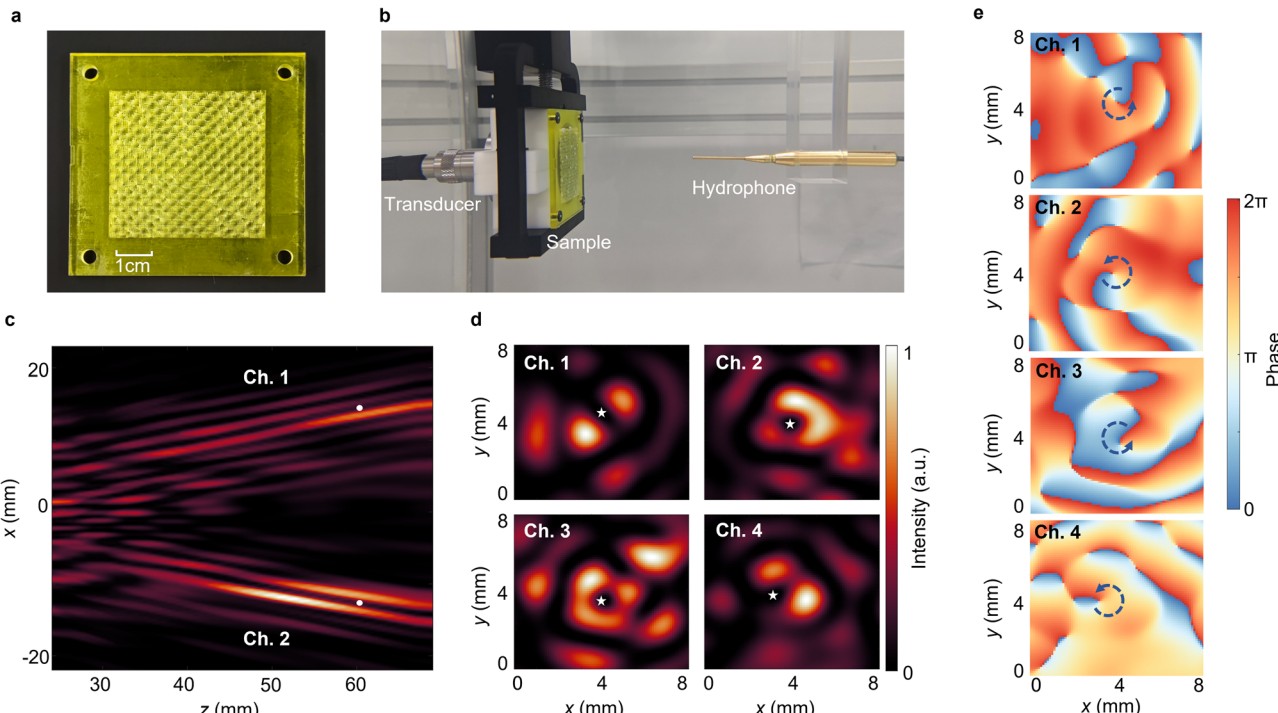

**Fig. 4 | Experimental results of +1-order sample in each direction. a** Sample manufactured by high-precision 3D printing. **b** The placement of transducers, sample, and hydrophones. **c**, **d** Intensity distribution measured experimentally in the *xoz* plane and *xoy* plane. **e** Phase distribution of four vortices.

with a step of 0.2 mm. The measured acoustic intensity distribution in the *yoz* plane (*x* = 0) is shown in Fig. 4c, illustrating the transition from overlapping to separated beams for two channels, which agrees well with the simulation results. In Fig. 4c, at *z* = 60 mm, the distance between the centers of the Channel 1 and Channel 2 beams (marked by white dots) measures

27.2 mm, yielding a calculated tilt angle of approximately 12.8° for the vortex beams, which matches the preset value. The local ultrasonic field near the vortices in the *xoy* plane (*z* = 62 mm) is also measured, and its intensity distribution is presented in Fig. 4d. Local minima at each vortex, representing the central phase singularities, are marked by stars. Figure 4e

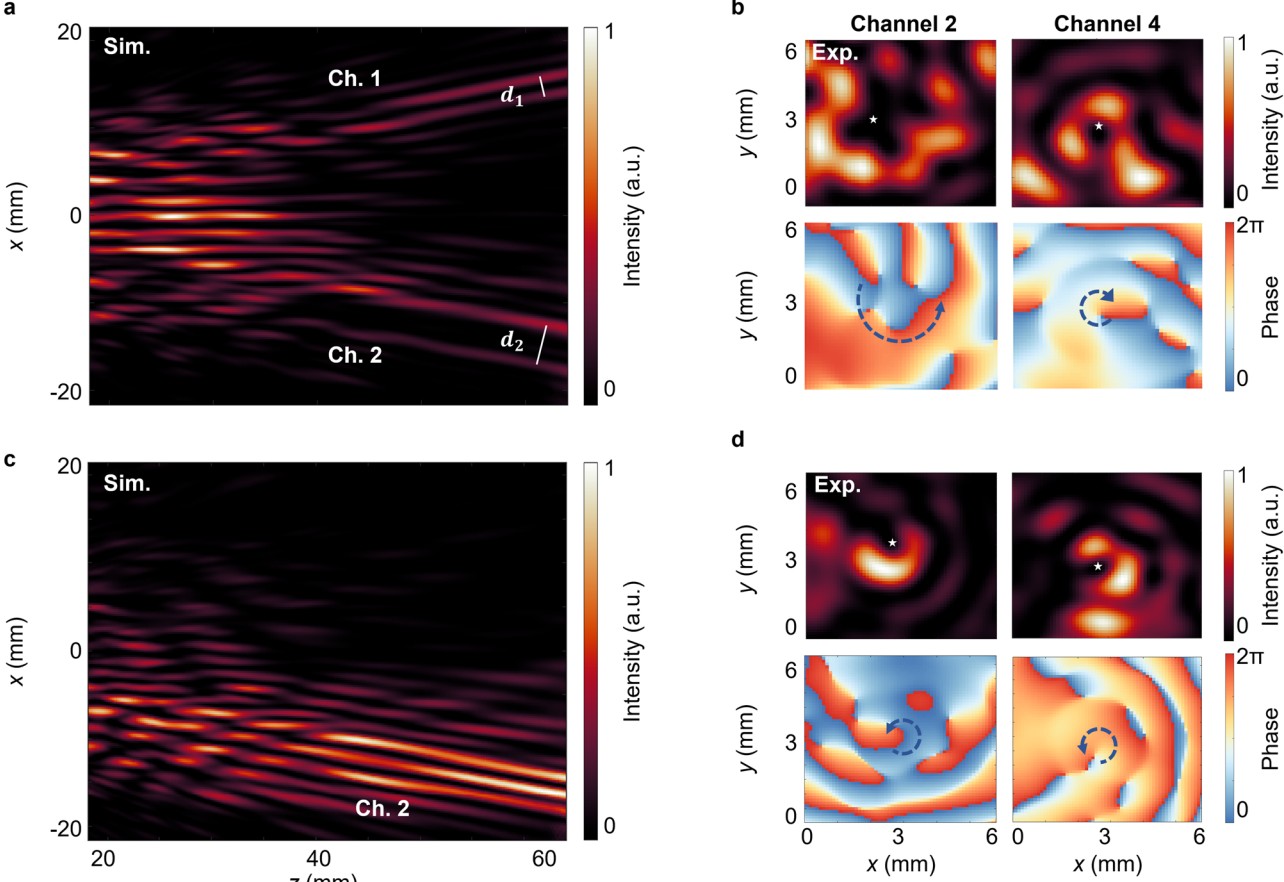

**Fig. 5 | Simulated and measured high-order vortex performance in two-channel configurations. a** Sound intensity distribution of high-order sample obtained by simulation in the *xoz* plane. **b** Experimental measurement of sound intensity and phase distributions for acoustic vortices in Channel 2 and 4 generated by the high-order sample. **c** Sound intensity distribution of two-channel sample obtained by simulation in the *xoz* plane. **d** Experimental measurement of sound intensity and phase distributions for acoustic vortices in Channel 2 and 4 generated by the two-channel sample.

displays the phase distribution of each channel, with arrows indicating the topological charge of m = +1 for each vortex. Our experimental results demonstrate the performance of the spatial multiplexing scheme and validate the feasibility of multi-channel ultrasonic vortices.

It is noteworthy that the proposed spatial multiplexing scheme offers an advantage in energy efficiency over the conventional phase superposition method. We conducted additional simulations (see Supplementary Note 3 for details), setting the tilt angle of all four beams to 6° to accentuate the differences. The results show that the main lobe energy ratios for the spatial multiplexing scheme (60.73%, 62.07%, 70.56%, and 48.27% for each channel) are generally higher than those of the phase superposition scheme (57.01%, 47.62%, 49.38%, and 48.02%). This proves that our scheme can more effectively concentrate energy in the main lobe, reduce sidelobe interference, and thereby offer higher energy utilization efficiency and clearer channel decoupling in practical applications.

### Adjustability of spatial multiplexing schemes

To further demonstrate the versatile functionalities of the proposed metalens, we modify the topological charges of two channels (Channel 2 and Channel 4) to $m_2 = +3$, $m_4 = -1$, while other channels remain unchanged. Figure 5a shows the simulated acoustic intensity distribution in the *xoz* plane ($y = 0$). The beam widths of the vortex beams from Channels 1 and 2, defined as the distance between two intensity maxima, are marked by white lines as $d_1 = 2$ mm, $d_2 = 4.9$ mm, respectively. The transverse size of the Channel 2 vortex beam is approximately 2.5 times larger than that of Channel 1, attributed to stronger phase variations near the central singularity of higher-order vortices, which enhances diffraction effects. Further

analysis was conducted on the far-field propagation characteristics of vortex beams of different orders to evaluate their energy retention. As presented in Supplementary Note 4 and Supplementary Fig. 5, the acoustic intensity of vortices with topological charges |m| = 1, 3, and 5 was tracked over a distance of 33 wavelengths. The results indicate intensity ratios of approximately 1: 0.72: 0.51, yet a consistent attenuation of about 40% across all orders, demonstrating similar propagation stability regardless of the topological charge. The sample is fabricated following the previously described method, and the corresponding experimental measurements are performed. The measured intensity and phase distributions of Channels 2 and 4 in the *xoy* plane ($z = 62$ mm) are presented in Fig. 5b. Higher-order vortices exhibit larger profiles. The phase distributions, indicated by arrows, confirm topological charges of +3 and −1 for the respective vortices.

The previously described metalens employs a four-channel multiplexing scheme based on the parity of both rows and columns. Here, we implement a two-channel multiplexing metalens by modifying the division protocol to depend solely on row parity: odd rows are assigned to Channel 2 and even rows to Channel 4. Figure 5c presents the simulated acoustic intensity distribution in the *xoz* plane ($y = 0$), revealing that the generated vortex beams exhibit higher intensity and superior contrast compared to the four-channel configuration. Figure 5d demonstrates the measured intensity and phase distributions in the *xoy* plane ($z = 62$ mm), which align with the predefined parameters, confirming the successful generation of +1-order Bessel vortices. For comprehensive validation, Supplementary Note 5 presents a full-wave numerical analysis of the transverse vortex profiles, showing excellent agreement with the experimental data in Fig. 5. Furthermore, we quantitatively computed the average acoustic intensity in the

vicinity of singularities for both two-channel and four -channel vortex configurations. The simulation results reveal that the two-channel scheme exhibits about 3.8 times greater acoustic intensity than its quad-channel counterpart. Supplementary Note 6 confirm that when distinct channels encode identical vortex beams, acoustic pressure fields become additively superimposed while intensity distributions exhibit quadratic enhancement. Although this reduces the number of channels, it enhances energy concentration and improves the signal-to-noise ratio.

## Quantitative modal analysis

The above experimental results qualitatively validate the generation of multi-channel vortices. To further quantitatively evaluate the modal purity of the generated vortex beams and the level of inter-channel crosstalk, we calculated the modulus of the normalized correlation coefficient between the simulated acoustic fields and the target ideal vortex modes (see Supplementary Note 7 for detailed methodology and results). For the +1-order metalens, the purities for the four channels are 82.18%, 86.37%, 85.14%, and 81.61%, respectively. For the dual-channel metalens, the purities for the two active channels are 80.36% and 76.32%. For the high-order metalens, the purities for the four channels are 85.33%, 82.65%, 76.99%, and 76.41%, respectively.

The quantitative modal purity analysis demonstrates that the spatially multiplexed metalens proposed in this work can generate ultrasonic vortex beams with predefined topological charges simultaneously or selectively with high fidelity. All calculated purity values are above 76%, indicating high mode generation efficiency and low inter-channel crosstalk. This provides key quantitative metrics for evaluating and comparing the performance of multi-functional vortex-generating devices.

## Potential errors and limitations

The excellent agreement between simulation and experiment confirms the effectiveness of our design. Nevertheless, potential influences from non-ideal factors require discussion. First, the fabrication tolerance of the 3D-printed metalens is approximately 10 μm. Given that this value is substantially smaller than the operational wavelength (0.75 mm), the resulting phase error is negligible compared to the full $2\pi$ phase span required for wavefront modulation. Consequently, its influence on the acoustic pressure distribution and phase profile remains minimal. Second, although the sample was rigidly mounted and aligned using a precision positioning system, minor installation deviations ( < 0.5 mm in position and <1° in tilt) are inevitable. Repeated measurements showed that the standard deviation of the vortex center position was less than 0.3 mm. Finally, the background noise in the tank environment was suppressed by averaging 32 acquisitions, resulting in a high signal-to-noise ratio (SNR > 30 dB), which minimally affects the phase and intensity measurements. Therefore, we conclude that these non-ideal factors are within acceptable limits and do not alter the main conclusions of this work.

## Discussion

We propose a MHz-band multiplexed metalens that encodes four-channel phase information into a single device through spatial multiplexing, while maintaining ultrasonic field fidelity. Unlike conventional phase superposition methods, this scheme can avoid the change in amplitude during phase superposition of multiple beams. Theoretical analysis validates the feasibility of this approach and evaluates its impact on ultrasonic field characteristics. Experimental and numerical studies demonstrate that the multiplexed metalens can simultaneously generate four +1-order Bessel vortices with distinct orientations underwater, achieving precise control over both topological charge and radiation direction with a directional deviation below 1° and a lateral spatial resolution on the order of the wavelength. The reliability of the multiplexed metalens in producing high-order vortices is further experimentally verified. By modifying the encoding protocol to regulate channel count, we developed a two-channel metalens exhibiting enhanced energy density with 3.8 times higher intensity.

Crucially, the spatial multiplexing architecture itself provides a fundamental advantage over static, single-function devices. While the channel count in this work was reconfigured by redesigning the metalens, the principle of interleaved pixels inherently allows for dynamic channel selection. Future implementations could achieve rapid, on-the-fly switching between channels—for instance, by employing a movable physical mask or an active spatial light modulator—to selectively activate desired pixel groups, a feature unattainable with traditional transducers or monolithic metasurfaces.

This work enables efficient and flexible excitation of underwater ultrasonic vortices, offering concrete pathways for applications such as high-throughput cell sorting within confined pathways by leveraging the size-dependent trapping of different-order vortices, or dynamic multi-channel information transmission through rapid channel switching, thereby increasing the bandwidth capacity of acoustic communication systems.

## Method
### Numerical simulation

The entire three-dimensional ultrasonic field is simulated using the k-Wave toolbox, which uses a k-space pseudo-spectral model to simulate the time domain of acoustic wave. The simulation domain consists of two materials: the metalens material and water. The density of water is 1000 kg m$^{-3}$, and its sound speed is 1500 m s$^{-1}$, while the density of the solid material is 1050 kg m$^{-3}$, with a sound speed of 2200 m s$^{-1}$, which is consistent with actual material properties. The outer layer of the entire domain is surrounded by a perfectly matched layer (PML) to ensure that no acoustic waves are reflected. The calculated time domain is sufficiently long to ensure that the entire ultrasonic field reaches a steady state.

### Sample manufacturing and experimental settings

The experimental sample is designed as a 60 mm × 60 mm square structure, comprising a 40 mm × 40 mm metalens region and a 10 mm-wide peripheral frame for mechanical fixation. The sample is fabricated via 3D printing technology using a photopolymer resin (HTL resin) with a printing resolution of 10μm. All measurements were conducted in a water-filled tank (90 × 60 × 60 cm³) integrated with a three-axis computer-controlled positioning system (UMS4, Precision Acoustics). The ultrasonic transducer (V395-SU, Olympus) was driven using an arbitrary waveform signal generator (33500B, Keysight Technologies) connected via a power amplifier (ATA-8202, Aigtek). The driving signal was a 30-cycle sinusoidal waveform with an amplitude of 316 mVrms, and the power amplifier was set to a gain of 27 dB. The sample is rigidly mounted at the transducer's focal plane to ensure perpendicular incidence. A calibrated hydrophone (NH1000, Precision Acoustics) was directly connected to a preamplifier and powered by a DC coupler, performing spatial field mapping with a scanning resolution of 0.2 mm across a predefined region to probe the ultrasound field. Time-domain signals were acquired using a digital oscilloscope (DSOX3024T, Keysight Technologies) with a sampling rate of 400 MHz and 32 averages.

## Data availability

The data are available from the corresponding authors upon reasonable request.

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

## Acknowledgements

This work is supported by the National Key R&D Program of China (Grant Nos. 2022YFA1404400 and 2022YFA1404403), the National Natural Science Foundation of China (Grant No. 92263208 and 12304494), the Research Grants Council of Hong Kong SAR (Grant AoE/P-502/20), the Fundamental Research Funds for the Central Universities.

## Author contributions

Z. Gu and J. Zhu conceived the study. Y. Su carried out the theoretical analysis and designed the experiments. Y. Su and D. Wang performed the measurement and the data processing. C. Liu contributed technical advice and ideas. Z. Gu and J. Zhu supervised the research. Y. Su and Z. Gu wrote the paper.

## Competing interests

The authors declare no competing interests.
