## [Transparent Peer Review file · Communications Engineering]

Multi-channel ultrasonic Bessel vortex beams by spatial multiplexing metalens

Corresponding Author: Professor Zhongming Gu

Version 0:

Reviewer comments:

Reviewer #1

(Remarks to the Author)

The manuscript by Su et al. presented a study in which the authors designed a spatially multiplexed metasurface capable of generating a beam that incorporates multiple Bessel vortex modes simultaneously. This represents a significant improvement compared to existing methods that typically generate and utilize a single mode, showing considerable potential for applications such as underwater acoustic communication, acoustic sensing, and manipulation of small targets. The paper is generally well-structured, but the following comments should be addressed to further improve its clarity, depth, and impact.

Major comments:

1. The authors assign modes using an “odd-column-odd-row, odd-column-even-row, even-column-odd-row, and even-column-even-row” scheme. In this method, each pixel contributes to two modes simultaneously due to its row and column parity. For example, an “odd-column-odd-row” pixel also participates in either “odd-column-even-row” or “even-column-odd-row” modes. This results in a lack of pixel independence between modes, resembling phase superposition schemes. It is recommended to clarify the actual pixel assignment strategy and explicitly distinguish this approach from traditional spatial multiplexing (where pixels are independent for each mode) and phase superposition (where all pixels contribute to all modes). Such clarification will help better highlight the novelty and logical consistency of the proposed design.
2. Since the core innovation of this work lies in the generation of a beam containing multiple Bessel vortex modes via a spatially multiplexed metasurface, a quantitative evaluation of the quality of these generated modes is essential. It is recommended that the authors provide a quantitative analysis, for instance, by calculating the mode purity, crosstalk between different modes, or the efficiency of mode generation. Such metrics are crucial for objectively assessing the performance of the proposed design and for comparing it with alternative approaches.
3. The authors should provide more in-depth discussion regarding the practical applications of this research. For instance, if applied to acoustic manipulation, what are the specific advantages of using beams with multiple vortex modes? If intended for communication purposes, it is important to address the challenge of high attenuation of MHz-frequency acoustic waves in water—how can the proposed system remain practical under such conditions? Including a concrete showcase application (e.g., a simplified simulation or experimental demonstration) in the manuscript would significantly strengthen the impact and applicability of the work.
4. The experimental system requires a more systematic description to ensure reproducibility and allow readers to fully follow the methodology. The authors should provide a schematic diagram illustrating the setup and operating principles of the system. For example, how is the arbitrary waveform generator used to drive the transducers? A clear visual representation of the system architecture would greatly improve readability and help readers better understand the signal generation and transmission process.

Reviewer #2

(Remarks to the Author)

This work presents a design method for underwater ultrasonic multi-channel Bessel vortex metalens based on a spatial multiplexing strategy. It enables the generation of multi-directional and multi-order acoustic vortex beams in the megahertz regime. By carefully designing the pixel allocation scheme, multiple channel information is encoded into a single metalens, achieving a compact structure and flexible control over the ultrasonic beam generation. The authors have conducted a systematic investigation combining theoretical modeling, numerical simulations, and experimental validations. The results

are clearly presented and demonstrate strong engineering potential and physical insights. The work shows a certain degree of originality and is suitable for publication after minor revisions.

1. Although the authors emphasize the distinction from conventional phase superposition methods, this comparison is only briefly mentioned in Supplementary Note 2. The difference remains vague in the main text. It is recommended to add a more explicit comparison, such as quantitative metrics or schematic illustrations, to clearly highlight the advantages of the proposed method in terms of energy conservation and directional decoupling.
2. When multiple channels are simultaneously activated, is there any cross-talk between them? Are the OAM modes still orthogonal? It is recommended to include an analysis of inter-channel orthogonality or provide mode purity evaluations using methods like Fourier-Bessel decomposition.
3. While the experimental results match the simulations well, the influence of non-ideal factors such as fabrication tolerances, installation deviations, and background noise is not addressed. A quantitative error analysis would improve the reliability of the experimental results.
4. Some references appear to be duplicated (e.g., References 36 and 38 are identical). It is suggested to eliminate redundant entries and provide a categorized summary of the main technical routes (e.g., active control, vortex-based communication, non-Hermitian manipulation) to improve the logical clarity of citations.
5. High-order acoustic vortices (e.g., $m = +3$) exhibit greater energy divergence in the experiments, which may limit their practical utility. It is recommended to provide a comparison of far-field energy distributions for different topological orders, or include attenuation curves of acoustic pressure amplitude to assess their energy retention and propagation stability.
6. Although the abstract and conclusion mention potential applications in particle manipulation and underwater communication, these aspects are still rather general. It is advised to specify concrete performance metrics (e.g., spatial resolution, beam steering range, bandwidth capacity) and clarify how this method outperforms existing approaches. Possible practical implementations (e.g., front-end modulators for acoustic arrays, multiplexed encoders for communication) should also be discussed.

Reviewer #3

(Remarks to the Author)

Ultrasonic vortex beams offer great potential in micro-particle manipulation and underwater communications. In this work, the authors realized multi-channel ultrasonic Bessel vortex beams via spatial multiplexing. Compared to previous works with fixed functionalities, this novel methodology possesses the flexible manipulations of ultrasonic beams, with the validation of both simulations and experiments. I suggest the acceptance of this paper if the following comments can be addressed.

1. The transducer and the vortex beam are in the shape of a circle, but the meta-lens is in the shape of a square. How did you handle the four corners?

2. What's the fabrication technology used for the sample? It's a critical challenge for meta-lens at megahertz and beyond.

3. It's suggested to add a discussion on potential schemes about the dynamic control of the meta-lens.

4. What are the limitations about the scheme of spatial multiplexing, e.g., the maximum channels.

5. It is not obvious that the phase distributions in Fig. 2a and 2b correspond to the colorbar below 2c. Please clarify

6. In fig.3, the intensity is given as being unitless from 0 to 1 but it was not stated that it was normalized

7. Fig 5. Why show simulation results over an x-z plane for a and c but experimental results over x-y planes for b and d?

8. There are many grammatical mistakes throughout the paper. Please improve the English. Here are some suggestions 'stemmed' in line 36 should be 'stemming'

A common mistake is either the absence of articles or a noun that is singular that should be plural. Some examples:

line 43-33 '... to study the generation of acoustic vortex...' in this case, making it plural would be better 'acoustic vortices' rather than adding an article 'the acoustic vortex.'

Figure 1a caption 'The Metalens converts plane wave...' There should be an 'a' or 'the' before plane wave, or make it plural. Also it doesn't make sense to capitalize metalens here. It's not capitalized in the main body of the text.

lines 89-91: "During axial propagation, such vortex exhibits $m \times 2\pi$ phase variation around the central phase singularity in transverse cross sectional plane." Both 'vortex' and 'cross-sectional plane' are article-less singular nouns here.

line 114 'using spatial multiplexing scheme'

line 129 'with slight portion'

line 165 'along predetermined direction'

Noun-verb agreement in lines 58-59 'beams' -> 'remains.' If beams is to be plural, then the verb should be 'remain.'

Figure 2b caption 'Pick out...' it sounds like a command to the reader, which is off-putting.

lines 144-147, some words are capitalized which should not be. This is also a common mistake throughout the paper.

line 156, there should be a comma between 'material' and 'respectively.'

Version 1:

Reviewer comments:

Reviewer #1

(Remarks to the Author)

The authors have addressed my comments properly. Congratulations to the authors for the beautiful work.

Reviewer #2

(Remarks to the Author)

The authors have answered my concern questions.

Reviewer #3

(Remarks to the Author)

i have no further comments

the authors have addressed my comments

Response Letter to Reviewer

We are grateful for the constructive comments on this manuscript (COMMS-25-0533-T) from the reviewers. In the text below, the comments are quoted in *blue italics* and followed by our response. We have also revised the manuscript accordingly, and these updates are highlighted in **red**.

Reviewer #1:

Comments:

The manuscript by Su et al. presented a study in which the authors designed a spatially multiplexed metasurface capable of generating a beam that incorporates multiple Bessel vortex modes simultaneously. This represents a significant improvement compared to existing methods that typically generate and utilize a single mode, showing considerable potential for applications such as underwater acoustic communication, acoustic sensing, and manipulation of small targets. The paper is generally well-structured, but the following comments should be addressed to further improve its clarity, depth, and impact.

Response:

We thank Reviewer #1 for the encouraging comments. Below we provide point-to-point response to reviewer's comments. We have also revised the manuscript following the reviewer's suggestions.

Comments:

1. The authors assign modes using an "odd-column-odd-row, odd-column-even-row, even-column-odd-row, and even-column-even-row" scheme. In this method, each pixel contributes to two modes simultaneously due to its row and column parity. For example, an "odd-column-odd-row" pixel also participates in either "odd-column-even-row" or "even-column-odd-row" modes. This results in a lack of pixel independence between modes, resembling phase superposition schemes. It is recommended to clarify the actual pixel assignment strategy and explicitly distinguish this approach from traditional spatial multiplexing (where pixels are independent for each mode) and phase superposition (where all pixels contribute to all modes). Such clarification will help better highlight the novelty and logical consistency of the proposed design.

Response:

We sincerely thank the reviewer for this insightful comment and for raising this important issue regarding the pixel assignment strategy. In our design, each pixel is treated as an independent point source, each with its own distinct phase value. According to the Huygens principle, the final overall acoustic field results from the linear superposition of the fields radiated by these

individual point sources. Each pixel belongs exclusively to one specific channel. For instance, a pixel located at an "odd-column-odd-row" position is assigned solely to Channel 1 and does not contribute to any other channel. Although the pixels from Channel 1 and those from Channel 2 together generate a complex wavefront near the metasurface, this complex pattern can essentially be viewed as the linear superposition of the acoustic fields generated by the two independent channels. Because we designed different propagation directions for the different channels, the acoustic fields of these channels eventually decouple in space as the sound propagates to the far field. The same principle applies to other adjacent channels.

The fundamental distinction between this approach and the phase superposition scheme (where all pixels contribute to all modes) lies in the fact that our scheme allows for dynamic control—for example, using a metal mask to block a specific channel and thereby interrupt its signal transmission. This kind of selective control is impossible to achieve with the phase superposition scheme.

Compared to traditional spatial multiplexing schemes, which tend to keep different signals spatially separated, our approach achieves spatial multiplexing at a subwavelength scale, or more precisely, employs an interleaving method for multiplexing. As a result, each channel benefits from a larger effective aperture, which enables the generated acoustic beams to exhibit superior directivity and reduced diffraction.

Comments:

2. Since the core innovation of this work lies in the generation of a beam containing multiple Bessel vortex modes via a spatially multiplexed metasurface, a quantitative evaluation of the quality of these generated modes is essential. It is recommended that the authors provide a quantitative analysis, for instance, by calculating the mode purity, crosstalk between different modes, or the efficiency of mode generation. Such metrics are crucial for objectively assessing the performance of the proposed design and for comparing it with alternative approaches.

Response:

Thank you for your valuable suggestion. We fully concur on the importance of quantitatively evaluating the quality of the multi-channel Bessel vortex beams generated by our spatially multiplexed metalens. In response, we have conducted a modal purity analysis for the three metalenses discussed in the main text (the uniform +1-order, dual-channel, and high-order configurations) based on simulated data and have added a detailed analysis.

The principle and formula for our modal purity calculation are as follows: Ideal Bessel acoustic vortex fields of orders ranging from $m = -3$ to $m = +3$ were generated as a set of orthogonal bases, according to Eq. (1) in the main text with $\alpha = 12^\circ$ and $\beta = \gamma = 0$. The modal purity for a specific charge was quantified by calculating the modulus of the normalized correlation

between the simulated acoustic pressure field distribution, $P(x, y)$, and the corresponding target ideal vortex mode, $\Psi_m(x, y)$. The specific formula used is:

$$\text{Purity}(m) = \left| \frac{\iint P(x, y) \cdot \Psi_m^*(x, y) dx dy}{\sqrt{\iint |P(x, y)|^2 dx dy} \cdot \sqrt{\iint |\Psi_m(x, y)|^2 dx dy}} \right|$$

where Ψ_m^* represents the complex conjugate of the ideal mode. This coefficient essentially measures the cosine similarity between the two fields, yielding a value between 0 and 1. A value closer to 1 indicates a higher degree of match between the generated field $P(x, y)$ and the target mode Ψ_m , implying higher modal purity and lower crosstalk with other modes.

The calculated results demonstrate high modal purity across all channels: For the +1-order metalens, the purities for the four channels are 82.18%, 86.37%, 85.14%, and 81.61%, respectively. For the dual-channel metalens, the purities for the two active channels are 80.36% and 76.32%. For the high-order metalens, the purities for the four channels are 85.33%, 82.65%, 76.99%, and 76.41%, respectively.

These quantitative metrics robustly confirm that our proposed spatially multiplexed metalens can efficiently generate vortex beams with high purity and low inter-channel crosstalk, faithfully matching the predefined topological charges. This analysis objectively assesses the performance of our design and validates its high fidelity for multi-functional ultrasound vortex manipulation. We believe these supplementary quantitative results provide clear performance benchmarks for the reviewer and readers, and will facilitate objective comparison with alternative approaches in the future.

The detailed analysis, including the calculation method and purity bar charts, has been incorporated into the Supplementary Information as a new, standalone Supplementary Note 7.

Supplementary Note 7. Quantitative Analysis of Vortex Beam Modal

Purity

To quantitatively evaluate the quality of the multi-channel Bessel vortex beams generated by our spatially multiplexed metalens, we calculated their modal purity. This method objectively reflects the mode generation efficiency and inter-channel crosstalk by measuring the degree of match between the generated acoustic field and the target ideal vortex mode.

According to Eq. (1) in the main text, setting $\alpha=12^\circ$, $\beta=\gamma=0$, we generated ideal Bessel acoustic vortex fields $\Psi_m(x, y)$ of orders from $m = -3$ to $m = +3$ as a set of orthogonal bases. The modal purity is calculated based on the modulus of the normalized correlation coefficient between the complex acoustic pressure distribution of the generated field, $P(x, y)$, and the target mode, $\Psi_m(x, y)$. The specific calculation formula is as follows:

$$\text{Purity}(m) = \left| \frac{\iint P(x,y) \cdot \Psi_m^*(x,y) dx dy}{\sqrt{\iint |P(x,y)|^2 dx dy} \cdot \sqrt{\iint |\Psi_m(x,y)|^2 dx dy}} \right|$$

where Ψ_m^* represents the complex conjugate of the ideal mode. This coefficient essentially calculates the cosine similarity between the two fields, with a value range of [0, 1]. A purity value closer to 1 indicates a higher degree of match between the generated field $P(x, y)$ and the target mode Ψ_m with topological charge m , meaning higher modal purity and lower crosstalk with other modes.

We analyzed the simulation results for the three metalenses discussed in the main text (uniform +1-order, dual-channel, and high-order) and calculated the modal purity of the vortex beam for each channel. The calculation results are presented as bar charts in Fig. S8. The modal purities for the four channels (all preset to $m = +1$) are 82.18%, 86.37%, 85.14%, and 81.61%, respectively. All channels exhibit high and consistent purity, confirming that the metalens can generate multiple high-quality, low-crosstalk +1-order vortices in parallel. The modal purities for the two active channels (both preset to $m = +1$) are 80.36% and 76.32%, respectively. The high purity levels demonstrate the effectiveness of the scheme under different channel configurations. The modal purities for the four channels (with preset topological charges of $m = +1, +3, +1, -1$, respectively) are 85.33%, 82.65%, 76.99%, and 76.41%. The high purity achieved for the higher-order vortices ($m = +3$ and $m = -1$) demonstrates the capability of the metalens to generate complex vortex modes.

Figure S8. Quantitative analysis of the modal purity of vortex beams generated by the metalens under different configurations. **a** Sample generating $m = +1$ vortices in all four channels. **b** Sample generating $m = +1$ vortices in two channels. **c** Sample generating higher-order vortices (Channel 2: $m = +3$, Channel 4: $m = -1$).

Comments:

3. The authors should provide more in-depth discussion regarding the practical applications of this research. For instance, if applied to acoustic manipulation, what are the specific advantages of using beams with multiple vortex modes? If intended for communication purposes, it is important to address the challenge of high attenuation of MHz-frequency

acoustic waves in water—how can the proposed system remain practical under such conditions? Including a concrete showcase application (e.g., a simplified simulation or experimental demonstration) in the manuscript would significantly strengthen the impact and applicability of the work.

Response:

We thank the reviewer for the valuable comments. We agree that a more in-depth discussion of practical applications is necessary. Regarding acoustic manipulation, we performed additional theoretical calculations to demonstrate the advantages of multiple vortex modes. Based on Gorkov theory, we calculated the acoustic radiation force potential (solid line) and the acoustic radiation force (dashed line) for both +1-order and +3-order vortex beams, as shown in the attached Figure R1. The results indicate that the potential well width for the +3-order vortex is approximately 4.5 mm, which is significantly wider than the 2.2 mm width for the +1-order vortex. This implies that higher-order vortex beams can trap larger particles (e.g., cells or large microparticles), and vortices of different orders can be employed for size-selective particle manipulation or sorting. This holds clear application potential in fields such as high-throughput cell analysis and micro-nano assembly.

Figure R1. Calculated acoustic potential well and radiation force distribution. Radial distributions of the normalized acoustic radiation force potential (solid line) and the acoustic radiation force (dashed line) for (a) +1-order and (b) +3-order ultrasonic vortex beams.

For communication applications, we acknowledge the challenge of high attenuation for MHz-frequency acoustic waves in water. However, the proposed spatial multiplexing scheme is frequency-adaptive and can function effectively at lower frequencies to mitigate attenuation. On the other hand, our system, leveraging multi-channel multiplexing, can achieve high-capacity data transmission over short distances (e.g., within underwater sensor networks or local communication scenarios). The orthogonality of multiple vortex modes can be exploited to encode multiple data streams, thereby improving spectral efficiency and partially compensating for the channel capacity limitations imposed by attenuation.

While the current manuscript focuses on validating the fundamental principle and performance of multi-channel vortex generation, we have provided quantitative analyses (e.g., modal purity in Supplementary Note 7) demonstrating low crosstalk and high reliability. The supplementary calculations (Fig. R1) further support its application prospects in particle manipulation. We plan to conduct more concrete application experiments in future work, such as dynamic particle manipulation or communication prototype demonstrations. We hope these additional discussions and data analyses strengthen the applicability and impact of our work.

Comments:

4. The experimental system requires a more systematic description to ensure reproducibility and allow readers to fully follow the methodology. The authors should provide a schematic diagram illustrating the setup and operating principles of the system. For example, how is the arbitrary waveform generator used to drive the transducers? A clear visual representation of the system architecture would greatly improve readability and help readers better understand the signal generation and transmission process.

Response:

We sincerely thank the reviewer for this valuable suggestion. We agree that a clearer and more systematic description of the experimental setup is essential for reproducibility and reader comprehension. In response to this comment, we provide a schematic diagram of experimental setup, as shown in Fig.R2, and we have revised the manuscript. In the Method section, “All measurements were conducted in a water-filled tank ($90 \times 60 \times 60 \text{ cm}^3$) integrated with a three-axis computer-controlled positioning system (UMS4, Precision Acoustics). **The ultrasonic transducer (V395-SU, Olympus) was driven using an arbitrary waveform signal generator (33500B, Keysight Technologies) connected via a power amplifier (ATA-8202, Aigtek). The**

driving signal was a 30-cycle sinusoidal waveform with an amplitude of 316 mVrms, and the power amplifier was set to a gain of 27 dB. The sample is rigidly mounted at the transducer's focal plane to ensure perpendicular incidence. A calibrated hydrophone (NH1000, Precision Acoustics) was directly connected to a preamplifier and powered by a DC coupler, performing spatial field mapping with a scanning resolution of 0.2 mm across a predefined region to probe the ultrasound field. Time-domain signals were acquired using a digital oscilloscope (DSOX3024T, Keysight Technologies) with a sampling rate of 400 MHz and 32 averages.”

Figure R2. The diagram of experimental setup.

Reviewer #2:

Comments:

This work presents a design method for underwater ultrasonic multi-channel Bessel vortex metalens based on a spatial multiplexing strategy. It enables the generation of multi-directional and multi-order acoustic vortex beams in the megahertz regime. By carefully designing the pixel allocation scheme, multiple channel information is encoded into a single metalens, achieving a compact structure and flexible control over the ultrasonic beam generation. The authors have conducted a systematic investigation combining theoretical modeling, numerical simulations, and experimental validations. The results are clearly presented and demonstrate strong engineering potential and physical insights. The work shows a certain degree of originality and is suitable for publication after minor revisions.

Response:

We thank Reviewer #2 for the encouraging comments. Below we provide point-to-point response to reviewer's comments. We have also revised the manuscript following the reviewer's suggestions.

Comments:

1. Although the authors emphasize the distinction from conventional phase superposition methods, this comparison is only briefly mentioned in Supplementary Note 2. The difference remains vague in the main text. It is recommended to add a more explicit comparison, such as quantitative metrics or schematic illustrations, to clearly highlight the advantages of the proposed method in terms of energy conservation and directional decoupling.

Response:

Thank you for your valuable suggestion. We fully agree on the need for a more explicit comparison between the spatial multiplexing scheme and the conventional phase superposition method. Accordingly, we have conducted additional 3D acoustic field simulations to compare the performance of the two methods in generating four-channel ultrasonic vortex beams. Specifically, the tilt angles of all four beams were set to 6° , under which the differences between the two schemes become particularly evident. We further calculated the proportion of the main lobe energy relative to the total beam energy (within a region extending 10 wavelengths around the main lobe) for each beam. The results show that the main lobe energy ratios for the spatial multiplexing scheme are 60.73%, 62.07%, 70.56%, and 48.27% for each channel, respectively, while those for the phase superposition scheme are 57.01%, 47.62%, 49.38%, and 48.02%. This demonstrates that the spatial multiplexing scheme can more effectively concentrate energy in the main lobes, reduce sidelobe interference, and thereby improve energy utilization efficiency and directional decoupling capability. We will add these quantitative results and corresponding schematic diagrams to the Supplementary Note 3 to more clearly illustrate the advantages of the proposed method:

Supplementary Note 3. Comparison of simulated acoustic fields between spatial multiplexing and phase superposition schemes.

To provide a quantitative comparison between the spatial multiplexing scheme and the conventional phase superposition method, we conducted additional 3D numerical simulations. Both schemes were configured to generate four vortex beams deflected at 6° relative to the z-axis. This specific tilt angle was chosen to make the differences in acoustic field distribution between the two schemes more pronounced.

The acoustic intensity distributions on the xoy plane for both schemes are presented in Fig. S4. The spatial multiplexing scheme produces cleaner and more distinct beam profiles with reduced spurious sidelobes compared to the phase superposition scheme.

To further quantify the performance, we calculated the energy concentration ratio for each beam, defined as the percentage of acoustic energy contained within the main lobe relative to the total energy in a surrounding region (within a 10-wavelength radius from each main lobe's center).

The main lobe energy ratios for the four channels generated by the spatial multiplexing scheme are 60.73%, 62.07%, 70.56%, and 48.27%, respectively. In contrast, the ratios for the phase superposition scheme are 57.01%, 47.62%, 49.38%, and 48.02%. The higher values achieved by the spatial multiplexing scheme demonstrate its superior capability in concentrating acoustic energy into the intended main lobes, thereby improving energy utilization efficiency and enhancing directional decoupling for multi-channel operation.

Figure S4. Comparison of simulated acoustic fields between spatial multiplexing and phase superposition schemes.

Comments:

2. When multiple channels are simultaneously activated, is there any cross-talk between them? Are the OAM modes still orthogonal? It is recommended to include an analysis of inter-channel orthogonality or provide mode purity evaluations using methods like Fourier-Bessel decomposition.

Response:

We sincerely appreciate the reviewer's important comment regarding the need for quantitative analysis of mode orthogonality and potential cross-talk during multi-channel operation. Addressing this point is crucial for evaluating the device performance. Following your suggestion, we have performed a modal purity evaluation for the three key metalens designs discussed in the manuscript (the uniform +1-order, dual-channel, and high-order vortex configurations).

The modal purity was calculated based on the modulus of the normalized correlation coefficient. Specifically, ideal Bessel vortex fields for orders $m = -3$ to $+3$ (generated according to Eq. (1) in the main text with $\alpha=12^\circ$, $\beta=\gamma=0$) served as the orthogonal basis set. The purity for a specific

topological charge m is defined by the following formula, comparing the simulated field $P(x, y)$ to the target mode $\Psi_m(x, y)$:

$$\text{Purity}(m) = \left| \frac{\iint P(x, y) \cdot \Psi_m^*(x, y) dx dy}{\sqrt{\iint |P(x, y)|^2 dx dy} \cdot \sqrt{\iint |\Psi_m(x, y)|^2 dx dy}} \right|$$

Here, Ψ_m^* denotes the complex conjugate of the ideal mode. This coefficient measures the cosine similarity between the two fields, yielding a value between 0 and 1. A value closer to 1 indicates higher modal purity and lower cross-talk with other modes.

The analysis results clearly demonstrate excellent performance: For the uniform +1-order sample, the purities for the four channels are 82.18%, 86.37%, 85.14%, and 81.61%, respectively. For the dual-channel sample, the purities for the two active channels are 80.36% and 76.32%. For the high-order sample, the purities for the four channels are 85.33% ($m = +1$), 82.65% ($m = +3$), 76.99% ($m = +1$), and 76.41% ($m = -1$), respectively.

All purity values are substantially high, providing strong quantitative support for our spatial multiplexing scheme: it can generate the predefined OAM modes with high fidelity and minimal inter-channel cross-talk when multiple channels are active, maintaining good mode orthogonality. We believe this supplementary analysis adequately addresses your concern.

The detailed analysis, including the calculation method and purity bar charts, has been incorporated into the Supplementary Information as a new, standalone Supplementary Note 7.

Supplementary Note 7. Quantitative Analysis of Vortex Beam Modal

Purity

In response to the reviewer's suggestion and to quantitatively evaluate the quality of the multi-channel Bessel vortex beams generated by our spatially multiplexed metalens, we calculated their modal purity. This method objectively reflects the mode generation efficiency and inter-channel crosstalk by measuring the degree of match between the generated acoustic field and the target ideal vortex mode.

According to Eq. (1) in the main text, setting $\alpha=12^\circ$, $\beta=\gamma=0$, we generated ideal Bessel acoustic vortex fields $\Psi_m(x, y)$ of orders from $m = -3$ to $m = +3$ as a set of orthogonal bases. The modal purity is calculated based on the modulus of the normalized correlation coefficient between the complex acoustic pressure distribution of the generated field, $P(x, y)$, and the target mode, $\Psi_m(x, y)$. The specific calculation formula is as follows:

$$\text{Purity}(m) = \left| \frac{\iint P(x, y) \cdot \Psi_m^*(x, y) dx dy}{\sqrt{\iint |P(x, y)|^2 dx dy} \cdot \sqrt{\iint |\Psi_m(x, y)|^2 dx dy}} \right|$$

where Ψ_m^* represents the complex conjugate of the ideal mode. This coefficient essentially

calculates the cosine similarity between the two fields, with a value range of [0, 1]. A purity value closer to 1 indicates a higher degree of match between the generated field $P(x, y)$ and the target mode Ψ_m with topological charge m , meaning higher modal purity and lower crosstalk with other modes.

We analyzed the simulation results for the three metalenses discussed in the main text (uniform +1-order, dual-channel, and high-order) and calculated the modal purity of the vortex beam for each channel. The calculation results are presented as bar charts in Fig. S8. The modal purities for the four channels (all preset to $m = +1$) are 82.18%, 86.37%, 85.14%, and 81.61%, respectively. All channels exhibit high and consistent purity, confirming that the metalens can generate multiple high-quality, low-crosstalk +1-order vortices in parallel. The modal purities for the two active channels (both preset to $m = +1$) are 80.36% and 76.32%, respectively. The high purity levels demonstrate the effectiveness of the scheme under different channel configurations. The modal purities for the four channels (with preset topological charges of $m = +1, +3, +1, -1$, respectively) are 85.33%, 82.65%, 76.99%, and 76.41%. The high purity achieved for the higher-order vortices ($m = +3$ and $m = -1$) demonstrates the capability of the metalens to generate complex vortex modes.

Figure S8. Quantitative analysis of the modal purity of vortex beams generated by the metalens under different configurations. a Sample generating $m = +1$ vortices in all four channels. **b** Sample generating $m = +1$ vortices in two channels. **c** Sample generating higher-order vortices (Channel 2: $m = +3$, Channel 4: $m = -1$).

Comments:

3. While the experimental results match the simulations well, the influence of non-ideal factors such as fabrication tolerances, installation deviations, and background noise is not addressed. A quantitative error analysis would improve the reliability of the experimental results.

Response:

We sincerely appreciate the reviewer’s insightful comments regarding the influence of non-ideal factors on our experimental results. We agree that a quantitative discussion of such errors significantly strengthens the reliability of the study. As suggested, we have now added a

dedicated subsection titled “Discussion on Potential Errors and Limitations” in the “Results and discussion” section to quantitatively address the influences of fabrication tolerances, installation deviations, and background noise.

Discussion on Potential Errors and Limitations

The excellent agreement between simulation and experiment confirms the effectiveness of our design. Nevertheless, potential influences from non-ideal factors require discussion. First, the fabrication tolerance of the 3D-printed metalens is approximately 10 μm . Given that this value is substantially smaller than the operational wavelength (0.75 mm), the resulting phase error is negligible compared to the full 2π phase span required for wavefront modulation. Consequently, its influence on the acoustic pressure distribution and phase profile remains minimal. Second, although the sample was rigidly mounted and aligned using a precision positioning system, minor installation deviations (< 0.5 mm in position and $< 1^\circ$ in tilt) are inevitable. Repeated measurements showed that the standard deviation of the vortex center position was less than 0.3 mm. Finally, the background noise in the tank environment was suppressed by averaging 32 acquisitions, resulting in a high signal-to-noise ratio (SNR > 30 dB), which minimally affects the phase and intensity measurements. Therefore, we conclude that these non-ideal factors are within acceptable limits and do not alter the main conclusions of this work.

Comments:

4. Some references appear to be duplicated (e.g., References 36 and 38 are identical). It is suggested to eliminate redundant entries and provide a categorized summary of the main technical routes (e.g., active control, vortex-based communication, non-Hermitian manipulation) to improve the logical clarity of citations.

Response:

Thank you for pointing out the error in the references. We have carefully checked the reference list and removed all duplicate entries. We corrected a mistakenly cited reference (the original reference [36] has been replaced with the correct citation, which now appears as reference [32] in the revised manuscript, ensuring accurate referencing throughout).

Furthermore, following the reviewer's suggestion to categorize and summarize the main technical routes, we have revised the Introduction section as follows:

“These methods can be broadly categorized into three technical routes based on their operational principles and flexibility. The first route relies on active transducer arrays^{30,31}, which offer dynamic programmability for vortex generation but at the cost of complex electronic control systems and high power consumption. The second route utilizes traditional passive components, such as spiral phase plates³² or optoacoustic conversion elements³³, which provide a compact and simple solution yet suffer from inherent limitations in functionality reconfiguration and single-operation mode. The third route is built upon acoustic metamaterials and metasurfaces, whose unique properties have transformed wavefront manipulation, enabling

precise control over acoustic fields at the subwavelength scale³⁴⁻⁴³. Notably, this route has further branched out into advanced design strategies, including topology-optimized metasurfaces for broadband vortex generation³⁸ and non-Hermitian metamaterials for switchable vortex emission^{39,40}. While metasurfaces have demonstrated remarkable capabilities in wavefront manipulation, such as focusing⁴⁴⁻⁴⁶ and holography⁴⁷⁻⁴⁹ in the MHz band, most existing passive designs are confined to generating a single vortex beam or a static multifunctional field. Consequently, the efficient generation of multichannel and multidirectional vortex beams in water via a single, passive device remains an open challenge, primarily due to the difficulty in decoupling multiple spatial information channels.”

Comments:

5. High-order acoustic vortices (e.g., $m = +3$) exhibit greater energy divergence in the experiments, which may limit their practical utility. It is recommended to provide a comparison of far-field energy distributions for different topological orders, or include attenuation curves of acoustic pressure amplitude to assess their energy retention and propagation stability.

Response:

Thank you for your valuable comment. We agree with your observation that high-order acoustic vortices (e.g., $m = +3$) exhibit greater energy divergence in the experiments. This phenomenon is inherently linked to the wave vector distribution of acoustic vortices. For a vortex beam carrying topological charge m , its wave vector (k_0) can be decomposed into a radial component (k_r) and an axial component (k_z), satisfying $k_0^2 = k_r^2 + k_z^2$. A higher-order vortex (larger m) possesses a steeper helical phase gradient, which necessitates a larger radial wave number (k_r) to support its phase variation. Consequently, this leads to a reduced axial wave number (k_z), thereby enhancing the beam's diffraction effect and causing it to spread more readily in the transverse plane, which manifests as greater energy divergence.

To quantitatively assess their potential for practical applications, we conducted further simulations focusing on their far-field energy distribution and propagation stability. We have supplemented our study with an investigation into the far-field energy distribution and acoustic pressure attenuation characteristics of vortices with different topological charges ($m = \pm 1, 3, 5$). The simulation results, as shown in the figure, present the variation curves of acoustic intensity with propagation distance for vortices of various orders over a range of $z = 71\text{--}96$ mm (exceeding 30 wavelengths).

The results indicate that the acoustic intensity of higher-order vortices is generally lower than that of lower-order ones. The intensity ratios among $|m| = 1, 3,$ and 5 vortices at the same position are approximately 1: 0.72: 0.51. However, more importantly, the acoustic intensity of vortices across all orders attenuates by about 40% over 33 wavelengths, demonstrating good propagation stability. Therefore, high-order acoustic vortices still hold potential for practical

applications that require larger orbital angular momentum, such as particle rotation or multi-channel communication.

We have added simulation results of the far-field intensity distribution of acoustic vortices with different topological charges:

Supplementary Note 4. Far-field intensity distribution of acoustic vortices with different topological charges

To quantitatively evaluate the intensity and attenuation of acoustic vortices of different orders, we conducted additional numerical simulations comparing the performance of vortices with topological charges $m = \pm 1, 3, 5$. We calculated the variation of acoustic intensity along the central axis of each vortex beam over a propagation distance from $z = 71$ mm to $z = 96$ mm (exceeding 30 wavelengths at 2 MHz).

The results are shown in Fig. S5. Higher-order vortices ($|m| = 3, 5$) have broader annular energy distribution profiles, with their acoustic energy distributed over a larger area, resulting in lower on-axis intensity compared to the fundamental $|m| = 1$ vortex. The intensity ratios of $|m| = 1, 3$, and 5 acoustic vortices at the same position are approximately 1:0.72:0.51, indicating that the vortex beam intensity is related to its topological order. Additionally, the intensity of vortices of all orders maintains 60% of its initial value after 25 mm of propagation, demonstrating good propagation stability of the acoustic beams.

Figure S5. Far-field intensity distribution of acoustic vortices with different topological charges. Acoustic intensity distribution curves along the propagation axis for vortices of orders $m = +1, +3, +5$. The intensity attenuation trends remain consistent for vortices of all orders over the propagation distance from 71 mm to 96 mm.

Comments:

6. Although the abstract and conclusion mention potential applications in particle manipulation and underwater communication, these aspects are still rather general. It is advised to specify concrete performance metrics (e.g., spatial resolution, beam steering range, bandwidth capacity) and clarify how this method outperforms existing approaches. Possible practical implementations (e.g., front-end modulators for acoustic arrays, multiplexed encoders for communication) should also be discussed.

Response:

The authors are grateful to the reviewer's valuable time on the manuscript. We agree that providing concrete performance metrics and a clearer discussion on advantages and practical implementations would strengthen our manuscript. We have now revised the Abstract and Discussion sections to address this point specifically.

In the Abstract part, "We experimentally designed a four-channel metalens with a high fabrication accuracy of 0.2 mm pixel size and measured the far-field ultrasound distribution in the water. Both topological charge and radiation direction of the generated vortices can be precisely controlled as predicted, showcasing great agreement with simulation results with a directional error of less than 1°."

In the Discussion part, "Experimental and numerical studies demonstrate that the multiplexed metalens can simultaneously generate four +1-order Bessel vortices with distinct orientations underwater, achieving precise control over both topological charge and radiation direction with a directional deviation below 1° and a lateral spatial resolution on the order of the wavelength. The reliability of the multiplexed metalens in producing high-order vortices is further experimentally verified. By modifying the encoding protocol to regulate channel count, we developed a two-channel metalens exhibiting enhanced energy density with 3.8 times higher intensity."

"Crucially, the spatial multiplexing architecture itself provides a fundamental advantage over static, single-function devices. While the channel count in this work was reconfigured by redesigning the metalens, the principle of interleaved pixels inherently allows for dynamic channel selection. Future implementations could achieve rapid, on-the-fly switching between channels—for instance, by employing a movable physical mask or an active spatial light modulator—to selectively activate desired pixel groups, a feature unattainable with traditional transducers or monolithic metasurfaces."

"This work enables efficient and flexible excitation of underwater ultrasonic vortices, offering concrete pathways for applications such as high-throughput cell sorting within confined pathways by leveraging the size-dependent trapping of different-order vortices, or dynamic multi-channel information transmission through rapid channel switching, thereby

increasing the bandwidth capacity of acoustic communication systems.”

Reviewer #3:

Comments:

Ultrasonic vortex beams offer great potential in micro-particle manipulation and underwater communications. In this work, the authors realized multi-channel ultrasonic Bessel vortex beams via spatial multiplexing. Compared to previous works with fixed functionalities, this novel methodology possesses the flexible manipulations of ultrasonic beams, with the validation of both simulations and experiments. I suggest the acceptance of this paper if the following comments can be addressed.

Response:

We thank Reviewer #3 for the encouraging comments. Below we provide point-to-point response to reviewer's comments. We have also revised the manuscript following the reviewer's suggestions.

Comments:

1. The transducer and the vortex beam are in the shape of a circle, but the meta-lens is in the shape of a square. How did you handle the four corners?

Response:

Thank you for raising this important question. The meta-lens sample is square because its parameters were saved in a matrix format during the sample design and simulation process. Although the transducer and the vortex beam are in the shape of a circle, and the transducer does not cover these four corners of the sample, the active area of the transducer is large enough to cover the vast majority of pixels. Therefore, the finally generated sound field is not significantly affected.

Comments:

2. What's the fabrication technology used for the sample? It's a critical challenge for meta-lens at megahertz and beyond.

Response:

Thank you for your insightful comment regarding the fabrication technology of the metalens. The samples were fabricated using Projection Micro Stereolithography (PμSL) technology via

a microArch S240 (BMF Precision Tech Inc.) high-precision 3D printing system, which provides a printing resolution of 10 μm . At the operational frequency of 2 MHz (wavelength ~ 0.75 mm in water), this fabrication precision is significantly finer than the wavelength, ensuring accurate phase modulation with negligible phase error from manufacturing tolerance. The excellent agreement between experimental and simulation results validates the feasibility of this fabrication technology for MHz-range acoustic metalenses.

Comments:

3. It's suggested to add a discussion on potential schemes about the dynamic control of the meta-lens.

Response:

We thank the reviewer for this valuable suggestion. We fully agree that a discussion on dynamic control schemes is crucial for enhancing the functionality and application potential of metasurfaces. In our manuscript, the spatial multiplexing metalens architecture we proposed, with its pixels interleaved according to parity of rows and columns, inherently provides a foundation for dynamic channel selection. Building on this architecture, a feasible dynamic control scheme could involve using a movable physical mask or an active spatial light modulator to selectively cover or activate specific pixel groups (e.g., all pixels in odd rows and odd columns for Channel 1), thereby enabling rapid switching between different vortex beam channels. This concept of "selective illumination" offers a path to dynamic control without refabricating the metalens itself. We have now revised the Discussion sections to address this point specifically.

In the Discussion part, “Crucially, the spatial multiplexing architecture itself provides a fundamental advantage over static, single-function devices. While the channel count in this work was reconfigured by redesigning the metalens, the principle of interleaved pixels inherently allows for dynamic channel selection. Future implementations could achieve rapid, on-the-fly switching between channels—for instance, by employing a movable physical mask or an active spatial light modulator—to selectively activate desired pixel groups, a feature unattainable with traditional transducers or monolithic metasurfaces.”

Comments:

4. What are the limitations about the scheme of spatial multiplexing, e.g., the maximum channels.

Response:

Thank you for raising this important question regarding the limitations of the spatial multiplexing scheme employed in our study. The primary limiting factor of this approach is the

inherent trade-off between the number of channels and the quality of the reconstructed acoustic field.

This trade-off arises fundamentally from spatial sampling principles: as the number of channels increases, the effective pixel size per channel also increases. According to the spatial sampling theorem, high-quality acoustic field reconstruction requires the equivalent pixel size per channel to be smaller than half the wavelength. The theoretical maximum number of channels, N_{max} can be estimated by:

$$a_{\text{eff}} = N \cdot a_0 \leq \frac{\lambda}{2}$$

where $a_0 = 0.2 \text{ mm}$ is the original pixel size, a_{eff} is the effective pixel size. For the current design, $N_{max} \approx 1.875$, implying a maximum of 2 channels per direction. We demonstrate that with a four-channel multiplexing configuration, each channel remains capable of covering most of the spatial frequencies within the cutoff wavenumber circle, with only minimal loss of high-frequency components. However, further increasing the number of channels beyond this level would likely compromise the reconstruction quality of the acoustic field.

Comments:

5. It is not obvious that the phase distributions in Fig. 2a and 2b correspond to the colorbar below 2c. Please clarify

Response:

Thank you for your valuable comment. We appreciate your observation and would like to clarify as follows:

The phase distributions in Fig. 2a, 2b, and 2c all share the same colorbar (located below Fig. 2c), which corresponds to a phase range of $[0, 2\pi]$. Fig. 2a and 2b illustrate the phase patterns of individual channels, while Fig. 2c shows the overall phase profile after spatial multiplexing. Therefore, the color mapping is consistent across all subfigures.

We have revised the details in the manuscript. In the caption of Fig. 1 part, “**All phase maps are plotted using the same colorbar ($[0, 2\pi]$).**”

Comments:

6. In fig.3, the intensity is given as being unitless from 0 to 1 but it was not stated that it was normalized

Response:

Thank you for your comment regarding Figure 3. We appreciate your observation regarding the unitless intensity values in Fig. 3, which were indeed presented in normalized form (ranging

from 0 to 1) to better illustrate the acoustic field distribution. We have now revised the figure caption to explicitly state "Intensity (a.u.)".

Figure 3 Simulation results of multi-channel acoustic vortex. a 3D cross-section of the entire sound field. **b** Directivity in the $y = 0$ plane. **c, d** Intensity distribution in xoz plane and xoy plane. **d** Phase distribution of each acoustic vortex.

Comments:

7. Fig 5. Why show simulation results over an $x-z$ plane for a and c but experimental results over $x-y$ planes for b and d?

Response:

Thank you for raising this important point. The $x-z$ plane views are optimal for illustrating the beam's behavior along the propagation direction (z -axis), including its designed tilt angle, non-diffracting characteristics, and the spatial separation of the multi-channel beams. However, high-resolution acoustic field mapping over the large spatial range required for $x-z$ planes is time-consuming. Therefore, we employed efficient numerical simulations to clearly and completely illustrate these propagation dynamics, as shown in Figs. 5a and 5c.

Experimentally, to concentrate on validating the practical performance and reliability of our design, we prioritized the measurement that most directly confirms the core vortex characteristics. We conducted high-precision scans in the $x-y$ plane at a fixed far-field distance ($z = 62$ mm). This plane is the most effective cross-section for observing the transverse intensity profile (doughnut shape) of the vortex and for measuring its helical phase to determine the

topological charge. The experimental data in subfigures (b) and (d) provide direct evidence of the successful generation of vortices with different orders ($m = +3$ and $m = -1$), demonstrating the reliability of our spatial multiplexing scheme in practical implementation.

We have added simulation results of the x - y plane in the supplementary materials for comparison with experimental results:

Supplementary Note 5. Simulated transverse profiles of multiplexed vortices

To provide a direct numerical counterpart to the experimental measurements, and to further validate our design, we performed additional simulations of the acoustic field in the transverse (xy) plane at the same far-field distance ($z = 62$ mm).

Figure S6 displays the simulated acoustic pressure and phase distributions for the two customized metalens designs. Figure S6a and b correspond to the two-channel sample (where only Channels 2 and 4 are active, both encoded with $m = +1$ vortices). The simulated donut-shaped pressure profile and the characteristic 2π -phase spiral in b confirm the successful generation of a high-quality $+1$ -order vortex. Figure S6c and d correspond to the high-order vortex sample (with Channels 2 and 4 encoded with $m = +3$ and $m = -1$, respectively, as in the main text). The simulated results clearly show the enlarged donut profile of the higher-order ($m = +3$) vortex and the opposite helicity of the phase for the negative-order ($m = -1$) vortex. These simulation results (Fig. S6) closely match the corresponding experimental measurements (Fig. 5b, d in the main text), confirming the successful generation of the predefined vortices.

Figure S6. Simulated acoustic field distributions in the transverse plane for customized

metalenses. a Acoustic pressure and **b** phase distributions generated by the two-channel metalens (Channels 2 & 4, both $m = +1$) at $z = 62$ mm. **c** Acoustic pressure and **d** phase distributions generated by the high-order metalens (Channel 2: $m = +3$, Channel 4: $m = -1$) at $z = 62$ mm.

Comments:

8. *There are many grammatical mistakes throughout the paper. Please improve the English. Here are some suggestions*

'stemmed' in line 36 should be 'stemming'

A common mistake is either the absence of articles or a noun that is singular that should be plural. Some examples:

line 43-33 '... to study the generation of acoustic vortex...' in this case, making it plural would be better 'acoustic vortices' rather than adding an article 'the acoustic vortex.'

Figure 1a caption 'The Metalens converts plane wave...' There should be an 'a' or 'the' before plane wave, or make it plural. Also it doesn't make sense to capitalize metalens here. It's not capitalized in the main body of the text.

lines 89-91: "During axial propagation, such vortex exhibits $m \times 2\pi$ phase variation around the central phase singularity in transverse cross sectional plane." Both 'vortex' and 'cross-sectional plane' are article-less singular nouns here.

line 114 'using spatial multiplexing scheme'

line 129 'with slight portion'

line 165 'along predetermined direction'

Noun-verb agreement in lines 58-59 'beams' -> 'remains.' If beams is to be plural, then the verb should be 'remain.'

Figure 2b caption 'Pick out...' it sounds like a command to the reader, which is off-putting.

lines 144-147, some words are capitalized which should not be. This is also a common mistake throughout the paper.

line 156, there should be a comma between 'material' and 'respectively.'

Response:

Thank you very much for your careful review and valuable suggestions. We sincerely appreciate your efforts in helping us improve the manuscript. We have carefully reviewed all the grammatical issues pointed out and have revised the entire manuscript accordingly. Specifically, we have addressed all the issues you mentioned:

“stemmed” in line 36 has been changed to “**stemming**”.

“acoustic vortex” in line 44 has been changed to “**acoustic vortices**”.

The caption of Figure 1a has been revised to “Figure 1 Design principles of spatial multiplexing **metalens**. a The **metalens** converts a plane wave into...”.

Added missing articles in lines 85–86: “such **a** vortex exhibits... in **the** transverse cross-sectional plane”.

Added “the” in “using **the** spatial multiplexing scheme” (line 117).

Revised “with slight portion” to “with **a** slight portion” (line 132).

Revised “along predetermined direction” to “along **a** predetermined direction” (line 170).

Revised the caption of Figure 2b from “Pick out...” to “**A zoom-in view of the phase distribution of Channel 4**”.

Corrected unnecessary capitalization in lines 144–147 and throughout the text.

Added a comma in line 156: “in the solid material, respectively”.